# Impact of Wedge Effect on Outcomes of Intertrochanteric Fractures Treated with Intramedullary Proximal Femoral Nail

**DOI:** 10.3390/jcm10215112

**Published:** 2021-10-30

**Authors:** Shen-Ho Yen, Cheng-Chang Lu, Cheng-Jung Ho, Hsuan-Ti Huang, Hung-Pin Tu, Je-Ken Chang, Chung-Hwan Chen, Sung-Yen Lin

**Affiliations:** 1Department of Orthopedics, Kaohsiung Medical University Hospital, Kaohsiung Medical University, Kaohsiung 80708, Taiwan; tennysonyen@gmail.com (S.-H.Y.); cclu0880330@gmail.com (C.-C.L.); Rick_free@mail2000.com.tw (C.-J.H.); hthuang@kmu.edu.tw (H.-T.H.); jkchang@kmu.edu.tw (J.-K.C.); hwan@kmu.edu.tw (C.-H.C.); 2Department of Orthopedics, School of Medicine, College of Medicine, Kaohsiung Medical University, Kaohsiung 80708, Taiwan; 3Orthopaedic Research Center, Kaohsiung Medical University, Kaohsiung 80708, Taiwan; 4Regeneration Medicine and Cell Therapy Research Center, Kaohsiung Medical University, Kaohsiung 80708, Taiwan; 5Department of Orthopedics, Kaohsiung Municipal Siaogang Hospital, Kaohsiung Medical University, Kaohsiung 80708, Taiwan; 6Department of Orthopedics, Kaohsiung Municipal Ta-Tung Hospital, Kaohsiung 80708, Taiwan; 7Department of Public Health and Environmental Medicine, School of Medicine, College of Medicine, Kaohsiung Medical University, Kaohsiung 80708, Taiwan; p915013@cc.kmu.edu.tw

**Keywords:** cut-out, femoral offset, intertrochanteric fracture, neck-shaft angle, wedge effect

## Abstract

The purpose of this study is to investigate the risk factors for wedge effect and its relevance between blade cut-out in patients with intertrochanteric fractures (ITF) treated with proximal femoral nail antirotation II (PFNA-II). A total of 113 patients with ITF treated with PFNA-II between 2012 and 2016 were retrospectively analyzed. Radiographic variables including preoperative fracture pattern, fracture classification, lateral wall fracture, and postoperative neck–shaft angle (NSA), femoral offset (FO), blade cut-out, and Parker’s ratio were measured for analysis. An average of 4.16° of varus malalignment in NSA and 5.5 mm of femoral shaft lateralization in FO was found post-operatively. The presence of lateral wall fracture was significantly related to post-operative varus change of NSA (*p* < 0.05). After at least one year of follow up, the blade cut-out rate was 2.65% (3/113), and Parker’s ratio was significantly higher in patients with blade cut-out (*p* = 0.0118). This study concluded that patients with ITF-present preoperative lateral wall fracture and postoperative higher Parker’s ratio in AP radiography showed higher incidence of wedge effect that might increase risk of blade cut out.

## 1. Introduction

Intertrochanteric fractures (ITF) frequently occur in the elderly, with an incidence that continues to rise as life expectancy increases [1]. ITF is mainly fixed with either extramedullary screw-plate devices such as dynamic hip screws (DHS) or intramedullary (IM) nails. DHS is considered the standard treatment for ITF and has been widely employed with long-lasting results. In recent decades, IM nailing became a popular option for ITF stabilization because of its mechanical advantage and rapid recovery from surgery compared with DHS [2]. The rate of IM nailing for ITF surprisingly increased from 3% in 1999 to 67% in 2006 in the USA [3]. Important issues concerned with increasing complications of proximal femur nail fixation have been recognized [4,5].

Several studies compared the clinical outcomes after treatment with DHS or IM nail for ITF; nonetheless, the results were inconclusive [6]. Because of its advantage, IM nailing is usually recommended for the fixation of unstable biomechanical ITF, as it reduces the lever distance to the weight-bearing axis [2]. However, IM nailing has reportedly higher rates of reoperation and complications than that of DHS, including implant cut-out through the femoral head, limiting its clinical safety [6]. Failed fixation of ITF would cause intractable pain and permanent functional impairment and further result in higher complications and poor prognosis. Conversion arthroplasty was the most reported intervention for this failure. However, conversional surgery is challenging because of soft tissue adhesion, residual anatomic alterations, and osteoporotic bone quality, consequently increased operation time, perioperative blood loss, iatrogenic fracture, and also a higher rate of dislocation [7,8,9].

Wedge effect, first described by O’Malley et al. [10], refers to the phenomenon in which femoral shaft lateralization and femoral neck varus malalignment occur following IM nail fixation for ITF (Figure 1 and Figure 2). The combination of varus malalignment and increased FO increases the bending force to the fracture site and implants, increasing the risk for implant cut-out after IM nailing [11,12]. Prior studies have investigated the risk factors for implant failure after fixation of the intramedullary nail. The center blade position, tip-apex distance (TAD) less than 25 mm, and good quality of fracture reduction should be attained to avoid implant failure [13,14]. The varus malalignment in wedge effect may increase the risk of improper reduction in the medial cortex and eccentric blade position linked to increased risk of implant failure.

Although wedge effect following IM nailing for ITF has been described, nevertheless, the associated risk factors for the incidence of wedge effect and its impact on clinic outcomes remain unclear. The present study aimed to investigate (1) the risk factors for wedge effect in patients with ITF treated with IM proximal femoral nail II (PFNA-II), and (2) the relation between wedge effect and blade cut-out failure after PFNA-II fixation. 

## 2. Materials and Methods

This study has been reviewed and approved by the Institutional Review Board of our hospital (registration number: KMUHIRB-E(I)-20180312). Patients sustaining unilateral ITF treated with PFNA-II between January 2012 and December 2016 were retrospectively analyzed. All surgery was operated with PFNA-II by the four hip surgeons who have more than five years surgical experiences to eliminate the bias of surgical technique and fixation implant. Patients who were followed up at least one year were included in the study. Conversely, patients with (1) pathological fractures, (2) previous contralateral hip fractures, (3) multiple fractures, (4) fractures extending to the subtrochanteric region, and (5) insufficient radiographic data after surgery which hindered accurate measurements, were excluded from the analysis. No patients were excluded based on age or comorbidities.

In order to survey the risk factors of wedge effect, radiographic variables including preoperative fracture pattern, fracture classification, presence of lateral wall fracture, and postoperative neck–shaft angle (NSA), femoral offset (FO), blade cut-out, and Parker’s ratio were measured for analysis.

Data were acquired from medical records and our hospital’s image database. Fractures were categorized into types in accordance with the AO Foundation and Orthopaedic Trauma Association (AO/OTA) fracture classification [15]. Stable ITF were defined as AO/OTA types 31-A1.1, 31-A1.2, 31-A1.3, and 31-A2.1, whereas unstable fractures were defined as AO/OTA types 31-A2.2, 31-A2.3, 31-A3.1, 31-A3.2, and 31-A3.3. Radiographs were reviewed by two well-trained hip surgeons, and radiographic variables were analyzed using our hospital’s picture archiving and communication system. Standardized postoperative anteroposterior (AP) radiographs of the hip were obtained with both legs internally rotated at 15°. Cross-table lateral radiographs were acquired with the contralateral hip flexed and abducted. Postoperative radiographs in the pelvic AP, hip AP, and lateral views were immediately performed after surgery before patients could leave the bed. Patients were scheduled for follow-up at six weeks, at three months, and subsequently every two months within one year. Preoperative and postoperative radiographs were evaluated and recorded for analysis by the first and second authors. None of the patients in this study were treated by these two authors.

Each patient underwent an X-ray examination to determine the integrity of the lateral wall. Neck-shaft angle (NSA), FO, and tip-apex distance (TAD) were evaluated on AP radiographs. NSA was defined as the intersection angle between the femoral neck axis and the femoral shaft axis (Figure 2) [16]. A change in NSA was measured by subtracting the NSA of the injured hip from that of the contralateral side. FO was defined as the perpendicular distance from the center of rotation of the femoral head to the long axis of the femoral shaft on AP radiographs (Figure 2) [17]. A change in FO was measured by subtracting the FO of the injured hip from that of the non-injured side. TAD and Parker’s ratio were used to evaluate the blade position. TAD was defined as the sum of the distance from the tip of the helical blade to the apex of the femoral head in AP and lateral views [18]. Parker’s ratio was measured according to the method described by Parker [19]. Using this method, the superior, inferior, anterior, and posterior borders of the femoral head were identified on AP or lateral radiographs. The blade position ratio was calculated by dividing the length from the blade’s center to the inferior border in AP view or posterior border in lateral view by the length of the femoral head in AP or lateral view, respectively.

### Statistical Analysis

Data were expressed as mean (standard deviation, SD), median (interquartile range, IQR), or counts and percentages. Characteristics of the study patients for the continuous and categorical variables were analyzed by Kruskal–Wallis test/Wilcoxon rank-sum test, and the chi-squared test/Fisher’s exact test, as appropriate, for comparisons between groups. Spearman’s Rank correlation coefficient was used to test the strength of the relationship between age, TAD, Parker’s ratio in AP and lateral, and outcome variables including the change of NSA and FO. This study also conducted the adjusted generalized linear regression model to analyze the relationship between the change of NSA or FO and variables including age, sex, fracture side, fracture classification, the subgroup with fracture stability, the subgroup with AO classification, lateral wall fracture, TAD, Parker’s ratio in AP and lateral, nail length, and blade cut-out. 

The adjusted odds ratio (OR) and 95% confidence interval (CI) were estimated by a stepwise logistic regression model. Potential risk factors were included in the analysis model, including age, sex, the fracture classification, the subgroup with fracture stability, lateral wall fracture, TAD, Parker’s ratio in AP and lateral, as well as NSA and FO difference. The correlation between changes in NSA and FO was further evaluated using the Pearson correlation coefficient. Statistical analysis was performed using SPSS version 20.0 (IBM Corp., Armonk, NY, USA) and SAS statistical package version 9.4 (SAS Institutes, Cary, NC, USA). *p* < 0.05 was considered statistically significant.

## 3. Results

A total of 162 consecutive patients (103 women, 59 men; mean age 79.7 years, range, 21–95 years) sustaining ITF who underwent surgical fixation using PFNA-II were assessed. Among these patients, 25 patients were excluded owing to inadequate follow-up and pathologic or multiple fractures, another 24 patients were further excluded because of a history of contralateral hip fracture. As a result, 113 patients (73 women, 40 men) with a mean age of 77.7 years (range, 21–95 years) met the inclusion criteria. The mean follow-up time was 17.3 months (range, 12–51 months). These patients all suffered from a fall from a standing height or minor traumatic events. Of all fractures, 17.7% (20/113), 65.5% (74/113), and 16.8% (29/113) were type 1, 2, and 3 fractures, respectively, according to AO/OTA classification. Furthermore, 25.7% (*n* = 29) of ITF occurred in the presence of lateral wall fracture on preoperative radiographs. 

### 3.1. The Decreased NSA and Increased FO after PFNA-II Fixation in ITF

Detailed demographic data and radiographic findings were presented in Table 1. The average blade position fell within the recommended optimal range, with the mean TAD being 22.7 mm and the mean Parker’s ratio on AP and lateral radiographs being 49.4% and 46.4%, respectively. The mean NSA was 128.9° for the operated hip and 133.1° for the contralateral non-injured hip. The mean FO of the injured side after fixation using PFNA-II was 5.5 mm longer than that of the non-injured side. 

### 3.2. The Presence of Lateral Wall Fracture and Parker’s Ratio on AP Radiography Are Risk Factors of Wedge Effect

This study investigated risk factors that may be associated with preoperative and postoperative changes in NSA and FO, and the results were presented in Table 2. No statistically significant differences in age, sex, fracture classification, nail length, and blade cut-out rate were observed about the changes in NSA and FO. However, this study noted a significant difference in the presence of lateral wall fracture and postoperative varus change in NSA. The change in NSA for ITF co-occurring with lateral wall fracture (average, −6.3°) was significantly more varus than that for ITF without lateral wall fracture (average, −3.4°; *p* = 0.042). Concerning the blade position, Parker’s ratio on AP radiographs was significantly associated with the change in NSA. Parker’s ratio on AP radiographs showed a moderate negative correlation with the change in NSA (Spearman rho, −0.37; *p* < 0.0001). Patients with greater varus change in NSA tended to have higher Parker’s ratio on AP radiographs. The correlation between changes in NSA and FO showed a negative correlation of −0.51 in Pearson’s correlation coefficient (Figure 3). For the correlation between changes in NSA and FO, the Pearson correlation coefficient was −0.51, indicating moderate positive correlation. Patients with more varus change in NSA tended to have a greater increase in FO.

### 3.3. Parker’s Ratio on AP Radiographs Was Highly Associated with the Occurrence of Blade Cut-Out

In this study, the blade cut-out rate was 2.65% (*n* = 3/113), and no significant differences in age, sex, fracture classification, presence of lateral wall fracture, nail length, TAD, and Parker’s ratio on lateral radiographs were observed with the blade cut-out. However, Parker’s ratio on AP radiographs was significantly higher in patients with blade cut-out than in those without blade cut-out (*p* = 0.0118; Table 3).

## 4. Discussion

The wedge effect is a common phenomenon among patients with ITF undergoing surgical fixation with proximal IM nail [10]. Whether the wedge effect may increase the mechanical failure rate of fixation with PFNA for ITF remains unclear. This study detected an average of 4.16° of varus malalignment and 5.5 mm of femoral shaft lateralization compared with those for the contralateral hip among patients with ITF treated with PFNA-II. Fracture pattern was not associated with the incidence of wedge effect; conversely, the co-occurrence of lateral wall fracture and Parker’s ratio on AP radiographs were correlated with postoperative varus malalignment. Furthermore, this study found that the postoperative Parker’s ratio on AP radiographs was significantly higher in patients with blade cut-out after IM nailing for ITF. 

Butler et al. reported that the wedge effect is mainly attributable to the difference in bone quality between the greater trochanter and the superolateral femoral neck [20]. The denser bone of the superolateral femoral neck may laterally divert reamers to the softer bone of the greater trochanter, resulting in a lateralized starting orifice. Inadequate reaming of the superolateral femoral neck may further block nail insertion, further leading to varus malalignment of the proximal fragments and femoral shaft lateralization. No reported study has investigated the risk factors for the wedge effect. It seems reasonable to assume that the difficulty in maintaining fracture stability during nailing for unstable ITF may influence the incidence or degree of wedge effect. However, our results showed that changes in NSA and FO between the operated hip and healthy contralateral side were not significantly associated with fracture type or stability. Consequently, this study did not observe a significant correlation between the incidence of wedge effect and fracture stability. As the wedge effect is related to the disparity in bone hardness around the starting orifice in the proximal femur, our study results identify that the presence of lateral wall fracture was significantly associated with postoperative varus malalignment. 

Postoperative varus malalignment and femoral shaft lateralization increase the mechanical load on implants, and a decrease in load on the bone may contribute to secondary varus collapse of the fracture or screw cut-out failure [11,12]. A slight valgus of the NSA, which increases the compression force at the interface of the fractured side, has been reported to reduce the risk of screw cut-out [21,22]. The biomechanical consequences of the wedge effect in patients with ITF remain unknown. Theoretically, the wedge effect after PFN fixation increases the mechanical load on implants. This study did not observe a significant association between blade cut-out failure and changes in NSA and FO. 

Numerous studies investigating the relationship between screw position and cut-out failure [5,23]. Superior positioning of the blade is a potential risk factor for blade cut-out failure in fixation with PFNA. Authors have suggested center-to-inferior positioning in the AP view and center-to-posterior positioning in the lateral view for the prevention of screw cut-out. This study demonstrated that varus change of NSA in wedge effect is significantly related to superior positioning of the blade that may increase the risk of blade cut-out. The result also showed that Parker’s ratio on AP radiographs was significantly higher in patients with blade cut-out. In spite of limited number of blade cut-out, this study assumed that the Parker’s ratio on AP radiographs would be a predictor of malposition of screw position, leading to cut-out risk.

In addition to the wedge effect, reduction loss in the lateral view after nail insertion may be another factor for superior positioning of the blade. Surgery for ITF is usually performed with the patient in a supine position on the fracture table. After nail insertion, the weight of the nail may draw down the femoral shaft and slightly increase the anteversion of the femoral neck. To solve this problem, operators usually increase the anteversion by rotating the nail. Rather than a cylinder shape, the PFNA-II has a medial-lateral angle of 5°. An increase in anteversion in the lateral view combined with superior positioning of the blade leads to a slightly higher Parker’s ratio on the hip AP view. 

This study has included patients aged less than 65 years treated by nail but not by plate, which might be a bias. According to a previous study, for stable ITF fractures and young patients, there is currently little evidence of the superiority between the DHS implant system and IM nails [2]. IM nails have better mechanical properties and proved to reduce postoperative pain and facilitate mobilization in unstable ITF [2,24]. In the present study, most ITF in patients aged younger than 65 years old were classified as unstable ITF (11/13) or with a lateral wall fracture (5/13). Considering the advantages in biomechanical properties and early mobilization, PFNA-II was undergoing for these patients in the study. Moreover, the results also demonstrated no significant correlation between age and wedge effect.

To our best knowledge, there are no studies that have investigated the risk factors for the wedge effect so far. The current study displayed the high incidence of wedge effect in nailing for ITF and also firstly indicated that the lateral wall fracture was significantly associated with postoperative varus malalignment. The lateral wall fracture has been considered as an unstable sign of ITF. The iatrogenic lateral damage while nailing should be prevented if possible.

The present study has some limitations. First, this study could not evaluate the preoperative NSA and FO of the injured hip. Despite the low probability, an asymmetry between the left and right femoral anatomy may occur among individuals. Second, the accuracy of NSA and FO measurements may be affected by hip position. External rotation of the femur should be avoided, as it may lead to an overestimation of NSA. Considering this, patients’ bilateral hips were maintained in internal rotation as much as possible during radiologic examination. Third, this study only analyzed the impact of the wedge effect in ITF treated with PFNA-II using radiographic findings. The wedge effect of other fixation devices to treat ITF cannot be concluded in this study. Fourth, only three cut-out cases were observed in this study, the effect of wedge effect on blade cut-out failure impact may not be conclusive. As the rate of blade cut-out failure in ITF treated with PFNA-II has been reported to be low, a larger case series is essential to clarify the impact of wedge effect on blade cut-out failure [25]. Finally, the impact of the wedge effect on functional outcomes was not discussed in this study. FO is an important factor to optimize muscle moment arms between the gluteus muscle and hip joint. The effects of increasing FO caused by the wedge effect may reduce the total hip abductor muscle strength. This hypothesis implies that further research should be conducted to evaluate the functional outcomes or postoperative gaits.

## 5. Conclusions

In conclusion, this study suggested that patients with ITF present preoperative lateral wall fracture, and postoperative higher Parker’s ratio on AP radiographs showed a higher incidence of wedge effect that might increase the risk of the blade cut-out failure after ITF fixation with PFNA-II.

## Figures and Tables

**Figure 1 jcm-10-05112-f001:**
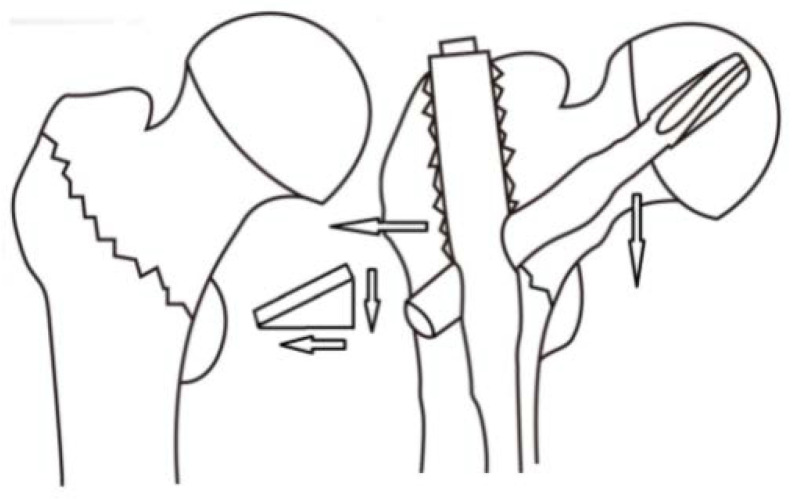
Schematic diagram illustrates the wedge effect including a varus deformity of the femoral neck and femoral shaft lateralization.

**Figure 2 jcm-10-05112-f002:**
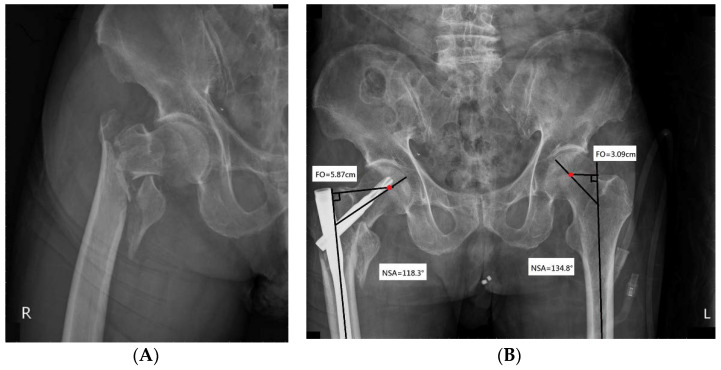
(**A**) An 85-year-old woman experienced severe hip pain after a fall. Radiographic examination of the right hip revealed an unstable ITF (type A2.2 according to AO/OTA classification). She underwent open reduction and internal fixation with PFNA-II. (**B**) Post-operative radiograph shows the wedge effect. The combination of decreased NSA and increased FO was noted upon comparison with the non-injured hip joint. Red dot represents the center of the femoral head.

**Figure 3 jcm-10-05112-f003:**
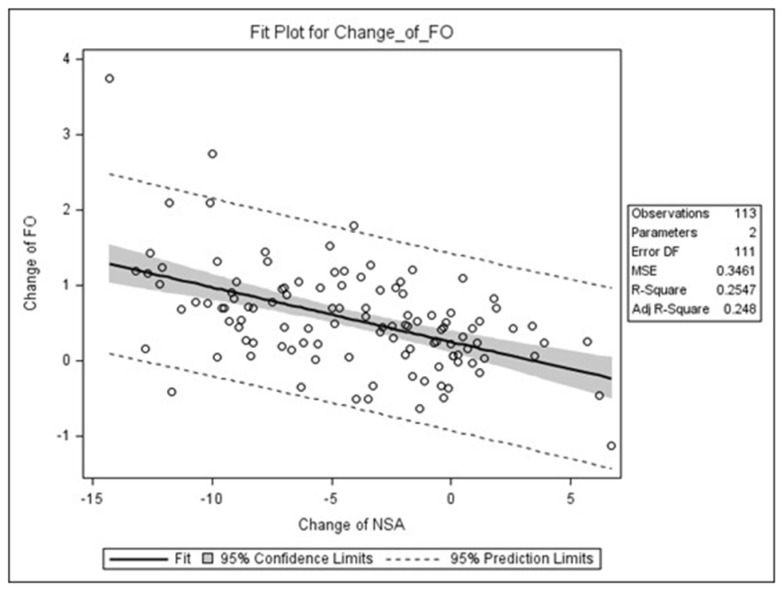
Scatter plot depicting the correlation between change of neck–shaft angle (NSA) and femoral offset (FO). There was a negative correlation between change of NSA and change of FO, Pearson correlation coefficient (r) = −0.51, *p* < 0.0001, with the change of NSA explaining 25.47% of the variation in change of FO.

**Table 1 jcm-10-05112-t001:** Patients’ demographics data and radiographic results.

	Patients
**N**	113
**Age (years), mean (SD)**	77.67 (12.11)
**Sex, n(%)**	
**Female**	73 (64.60)
**Male**	40 (35.40)
**Fracture side, n(%)**	
**Left**	54 (47.79)
**Right**	59 (52.21)
**Fracture classification, n(%)**	
**11**	1 (0.88)
**12**	14 (12.39)
**13**	5 (4.42)
**21**	40 (35.4)
**22**	22 (19.47)
**23**	12 (10.62)
**31**	4 (3.54)
**32**	7 (6.19)
**33**	8 (7.08)
**Lateral wall fracture, n(%)**	
**No**	84 (74.34)
**Yes**	29 (25.66)
**TAD (cm), mean (SD)**	2.27 (0.63)
**Parker’s ratio in AP(%), mean (SD)**	49.38 (7.96)
**Parker’s ratio in Lat(%), mean (SD)**	46.44 (8.59)
**NSA (°), mean (SD)**	
**Non-injury (n = 113)**	133.07 (4.41)
**Injury (n = 113)**	128.90 (5.27)
**Difference of SN angle (n = 113)**	−4.16 (4.73)
***p* value for Wilcoxon-Signed-Rank test**	<0.0001
**FO (cm), mean (SD)**	
**Non-injury (n = 113)**	6.80 (0.73)
**Injury (n = 113)**	7.35 (0.77)
**Difference of offset (n = 113)**	0.55 (0.68)
***p* value for Wilcoxon-Signed-Rank test**	<0.0001
**Nail length, n(%)**	
**170 cm**	6 (5.31)
**200 cm**	23 (20.35)
**240 cm**	48 (42.48)
**Long nail**	36 (31.86)
**Blade cut-out, n (%)**	
**No**	110 (97.35)
**Yes**	3 (2.65)

**Table 2 jcm-10-05112-t002:** Possible factors associated with wedge effect.

	N	Change of NSA	*p* Value	Adjusted P	Change of FO	*p* Value	Adjusted P
**Age, Spearman’s rho †**	113	0.09	0.364	0.1048	−0.1	0.3058	0.2191
**Sex, median (IQR)**							
**Female**	73	−3.50 (−7.80, −0.20)			0.45 (0.08, 0.76)		
**Male**	40	−4.75 (−8.85, −1.60)	0.2229	0.6532	0.62 (0.26, 1.01)	0.0951	0.0700
**Fracture side, median (IQR)**							
**Left**	54	−3.60 (−7.1, −0.10)			0.42 (0.07, 0.83)		
**Right**	59	−4.10 (−8.90, −0.50)	0.5310	0.3084	0.54 (0.23, 1.02)	0.1100	0.0418
**Fracture classification, median (IQR)**							
**11**	1	−0.20 (−0.20, −0.20)			0.50 (0.50, 0.50)		
**12**	14	−4.15 (−7.10, −2.10)			0.69 (0.15, 1.04)		
**13**	5	−5.00 (−10.00, −1.40)			0.70 (0.53, 1.16)		
**21**	40	−3.45 (−6.60, −0.25)			0.51 (0.22, 0.97)		
**22**	22	−2.05 (−6.90, 0.10)			0.46 (0.24, 0.87)		
**23**	12	−8.10 (−9.30, −3.95)			0.51 (0.06, 0.77)		
**31**	4	−7.05 (−8.70, −2.50)			0.12 (0.05, 0.61)		
**32**	7	−0.70 (−7.10, 0.10)			0.19 (−0.37, 0.91)		
**33**	8	−4.30 (−9.10, −0.45)	0.5225	0.9187	0.55 (0.02, 0.78)	0.8076	0.5030
**Subgroup with fracture stability**							
**Stable fracture type (11, 12, 13, 21)**	60	−3.70 (−7.05, −0.65)			0.53 (0.22, 1.01)		
**Unstable fracture type (22, 23, 31, 32, 33)**	53	−3.60 (−8.80, −0.30)	0.7041	0.7338	0.45 (0.07, 0.82)	0.2884	0.6887
**Subgroup with AO classification**							
**1 (11,12,13)**	20	−4.15 (−7.70, −1.55)			0.69 (0.20, 1.04)		
**2 (21,22,23)**	74	−3.55 (−8.30, −0.40)			0.47 (0.22, 0.89)		
**3 (31,32,33)**	19	−3.00 (−8.40, −0.10)	0.9016	0.9158	0.23 (0.02, 0.78)	0.2082	0.6213
**Lateral wall fracture, median (IQR)**							
**No**	84	−3.35 (−7.00, −0.15)			0.48 (0.15, 0.90)		
**Yes**	29	−6.30 (−9.80, −0.60)	0.0422	0.0919	0.48 (0.15, 1.02)	0.6081	0.8448
**TAD, Spearman’s rho †**		−0.1	0.2845	0.2289	0.18	0.0590	0.0793
**Parker’s ratio in AP%, Spearman’s rho †**		−0.37	<0.0001	<0.0001	0.09	0.3584	0.0158
**Parker’s ratio in lateral %, Spearman’s rho †**		−0.09	0.3355	0.4066	0.1	0.2904	0.9513
**Nail length, median (IQR)**							
**170 mm**	6	−1.55 (−3.60, −0.20)			0.65 (0.50, 0.96)		
**200 mm**	23	−5.00 (−7.70, 0.10)			0.43 (0.14, 0.91)		
**240 mm**	48	−3.80 (−8.55, −0.20)			0.56 (0.12, 1.04)		
**Long nail**	36	−4.10 (−8.75, −0.50)	0.6938	0.4790	0.45 (0.12, 0.75)	0.4783	0.1263
**Blade cut-out, median (IQR)**							
**No**	110	−3.6 (−7.80, −0.30)			0.49 (0.14, 0.95)		
**Yes**	3	−8.9 (−9.30, −0.40)	0.4267	0.5937	0.45 (0.41, 0.52)	0.8373	0.5878

NSA: neck-shaft angle; FO: femoral offset; TAD: tip apex distance. Data presented as median (interquartile range, IQR). Change of NSA or change of FO dependent variable was analyzed by using the Kruskal–Wallis test for comparisons between two or more groups of an independent variable. † Spearman’s rho (correlation coefficient) was calculated using Spearman’s rank correlation. The adjusted P was estimated by using a generalized regression model.

**Table 3 jcm-10-05112-t003:** Comparison of patient characteristics and technical variables in patients with blade cut-out and fracture union.

	No Blade Cut-Out Group	Blade Cut-Out Group	*p*	Adjusted OR (95%CI)	*p*
**N**	110	3			
**Age years, median (IQR)**	81.00 (74.00, 85.00)	78.00 (77.00, 81.00)	0.6230	-	-
**Sex**					
**Female**	70 (63.64)	3 (100.00)			
**Male**	40 (36.36)	0 (0.00)	0.5510	-	-
**Fracture classification, n(%)**					
**Subgroup with fracture stability**					
**Stable (11, 12, 13, 21)**	59 (53.64)	1 (33.33)			
**Unstable (22, 23, 31, 32, 33)**	51 (46.36)	2 (66.67)	0.5993	-	-
**Subgroup with AO classification**					
**1 (11,12,13)**	20 (18.18)	0 (0.00)			
**2 (21,22,23)**	71 (64.55)	3 (100.00)			
**3 (31,32,33)**	19 (17.27)	0 (0.00)	1.0000	-	-
**Lateral wall fracture, n(%)**					
**No**	83 (75.45)	1 (33.33)			
**Yes**	27 (24.55)	2 (66.67)	0.1613	-	-
**TAD, median (IQR)**	2.22 (1.87, 2.62)	2.52 (1.94, 2.68)	0.6617	-	-
**Parker’s ratio in AP%, median (IQR)**	48.27 (43.83, 52.95)	60.82 (59.1, 66.67)	0.0118	1.20 (1.03–1.40)	0.0181
**Parker’s ratio in lateral%, median (IQR)**	46.83 (39.73, 52.82)	47.66 (47.19, 52.44)	0.5921	-	-
**NS angle difference, median (IQR)**	−3.60 (−7.80, −0.30)	−8.90(−9.30, −0.40)	0.4267	-	-
**FO difference, median (IQR)**	0.49 (0.14, 0.95)	0.45(0.41, 0.52)	0.8373	-	-

Standard deviation (SD), median (interquartile range, IQR). NSA: neck-shaft angle; FO: femoral offset; TAD: tip apex distance. *p*-value was calculated using Wilcoxon Rank Sum test or chi-squared test/Fisher exact test, as appropriate, for comparisons between Fracture union group and Blade cut-out group. Data presented as mean (standard deviation, SD) or proportion. The adjusted OR and 95% CI were estimated by a stepwise logistic regression method; the significant variables were entered to this model (*p* < 0.05).

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
