# Peer review of "Impact of Wedge Effect on Outcomes of Intertrochanteric Fractures Treated with Intramedullary Proximal Femoral Nail"

_jcm, 2021, doi:10.3390/jcm10215112_

Round 1

Reviewer 1 Report

Review

Many thanks to the authors for having presented a so interesting retrospective study about “Impact of wedge effect on outcomes of intertrochanteric fractures treated with intramedullary proximal femoral nail”.

Before resubmitting the revision version of the article, please read the editorial rules carefully, and check other editorial aspects, such as: text alignment (lacking), text justification at the head (lacking), etc.

The language is good, hence the manuscript does not need to be corrected by a person of English mother tongue.

Abstract

The abstract stands alone and it captures the appropriate essence of the manuscript. It is well structured, and it contains the main information of the study.

Key words

Please provide the key words in alphabetic order.

Background

The introduction identifies the problem that is being address in the manuscript. It clearly develops and states the purpose of the manuscript, identifying the problem and clearly stating the purpose of the study. However, is toot focus on pertrochanteric fractures. What’s about the proximal femoral fractures treated with screws?

Line 44:  At the same time, important issues concerned with increasing complications of proximal femur nail fixation have been recognized.

Please compare the experience with intertrochaneric fractures with those of proximal femour treated by screws, quoting:

Predictors of early failure of the cannulated screw system in patients, 65 years and older, with non-displaced femoral neck fractures. Aging Clin Exp Res. 2020 Mar;32(3):505-513. doi: 10.1007/s40520-019-01394-1. Epub 2019 Nov 1.

Methods

This section contains enough information to understand and possibly repeat the study. The experimental design sounds good. There is an appropriate number of patients to justify the results. However, it does not reflect the Strobe Statement-Checklist for cohort studies.

Line 76: as it contains some results, they must move into Results section, including characteristics of the sample.

Line 75: please provide the registration number of the study by the Institutional Review Board and year of approbation (also here).

Line 97: please report who evaluated preoperative and postoperative radiographs and recorded the data for analysis.

Statistical analysis

Line 113: please provide who performed the analysis: an independent statistician or the same authors?

Results

The results presented are quite complete, reflecting the MM section. Please add here the results provide in MM section.

The results are reproducible and reflective of clinical expectations. They are displayed in a readable fashion.

However, the main bias of this study is having included patient less then 65yrs treated by nail and not by plate. Please discuss this aspect in the discussion section, justifying your indications and the rationale of your clinical choice. Also compare your results with others of proximal femoral fractures quoting:

Management and treatment of femoral neck stress fractures in recreational runners: a report of four cases and review of the literature PMID: 29083360 PMCID: PMC6357658 DOI: 10.23750/abm.v88i4-S.6800

Discussion

The length and content of the discussion communicates the main information of the paper. The discussion explains the results relative to prior publication. However, it does recognize the limitations of the manuscript. Please provide a separate paragraph describing strength and limits of the study.

Conclusion

The conclusions only reflect and refer to the results of the study, being justified by the results and the methods.

Please in the entire manuscript do not use the first plural person (We), but the impersonal form.

Line 267: In conclusion, we suggested that patients…Better: In conclusion, this study suggested that patients….

References

The references are up to date, but they should be integrated as suggested. Some references are not up to date. Hence, delate those before 2010 if not essential, replacing them with newer ones and integrate them as suggested previously.

Tables and Figures

The number and quality of tables and figures are appropriate to transmit the main information of the paper.

Reviewer 2 Report

This is a good  review about intertrochanteric fractures.

You said, Pearson correlation coefficient, -0.51, I think correlation rate is evaluate square, this square 0.51×0.51 , about .025, .

Over correlation square rate >0.40 is evaluated, this score , 0.25 is inadequate.

Reviewer 3 Report

Introduction

In addition to the brief overview, please add clinical characteristics of implant failure (increased reoperation rate, infection rate, healthcare costs).

Please add the tip-apex distance as prognostic parameter for femoral cut-off after ITF.

Fig. 1 and 2: Very good explanatory x-rays. Maybe you could improve them with adding the respective plain and marking the center of femoral head (i.e. in red).

Materials and Methods

How many "experienced surgeons" operated? Did they reliably use the same technique?

How did you make sure the data was normally distributed?

Did you perform any power analysis before to make sure you got the needed statistical power?

Why did you assume normality and choose pearsons correlation instead of spearman?

Results

What were the reasons of the ITF fractures? They may all have been traumatic, but due to fall from height or minor/major traumatic events?

An offset difference of FO between the operated and non-operated side is not a surprise. So the statistical evaluation might as well be deleted.

The term "lateral wall fracture" should not be introduced in the results section, but rather as an assessed parameter in the "methods" section. This way it does not seem as if you were just looking for parameters which might be associated with changes in NSA and FO.

How exactly were the "factors associated with wedge effect" calculated?

It has to be clearly stated that the n=3 in the "blade cut-out group" is a limitation.

Please perform a multivariate analysis regarding possible influences of age, gender, nail length, ... on FO, NSA and the possibility of blace cut-out.

Why is the presence of a lateral wall fracture no predictor for blade cut-out in table 3?

Discussion

line 240-241: "needs to be further evaluated".

Round 2

Reviewer 1 Report

The paper did not improve significantly respect to the previous version, as the authors ignored most of my suggestions.

When the authors will provide a serious revision of the manuscript, I will happy to consider it for further revision or potential acceptation.

Reviewer 2 Report

This paper is well described.

However this paper is not valid to read, the reason why the main reason of cut out is due to the position of lag screw.
